# Mutation Spectrum of *GAA* Gene in Pompe Disease: Current Knowledge and Results of an Italian Study

**DOI:** 10.3390/ijms25179139

**Published:** 2024-08-23

**Authors:** Marta Moschetti, Alessia Lo Curto, Miriam Giacomarra, Daniele Francofonte, Carmela Zizzo, Elisa Messina, Giovanni Duro, Paolo Colomba

**Affiliations:** Institute for Biomedical Research and Innovation (IRIB), National Research Council (CNR), 90146 Palermo, Italy; marta.moschetti@irib.cnr.it (M.M.); alessia.l_86@hotmail.it (A.L.C.); miriam.giacomarra@irib.cnr.it (M.G.); daniele.francofonte@irib.cnr.it (D.F.); carmela.zizzo@irib.cnr.it (C.Z.); elisa.messina@irib.cnr.it (E.M.); giovanni.duro@irib.cnr.it (G.D.)

**Keywords:** Pompe disease, misdiagnosis, metabolic myopathy, therapeutic procedures, mutation spectrum

## Abstract

Studying a patient with Pompe disease (PD) is like opening Pandora’s box. The specialist is faced with numerous clinical features similar to those of several diseases, and very often the symptoms are well hidden and none is associated with this rare disease. In recent years, scientific interest in this disease has been growing more and more, but still no symptom is recognized as key to a correct diagnosis of it, nor is there any specific disease marker to date. New diagnostic/therapeutic proposals on disease allow for the diffusion of knowledge of this pathology for timely diagnosis of the patient. Due to unawareness and difficulty in diagnosis, many adults with PD are diagnosed with great delay. In this article, we report and discuss current knowledge of PD and provide new data from work conducted on a cohort of 2934 Italian subjects recruited in recent years. A genetic analysis of the *GAA* gene was performed on patients with significant clinical signs and pathological enzyme activity to define the genetic profile of subjects. This identified 39 symptomatic PD subjects with low acid alpha-glucosidase enzyme activity and the presence of two causative mutations in *GAA* gene regions. Furthermore, 22 subjects with genetic variants of uncertain significance (GVUS) were identified.

## 1. Introduction 

The lives of patients and families facing the challenges of rare and/or severely disabling diseases have changed compared to 20 years ago. Clinicians spend a lot of time studying their symptoms and trying to understand how to approach patients’ problems.

But sometimes that is not so easy. Time has a significant impact on clinician’s efforts and attention. This restriction could result in the clinician identifying the disease incorrectly and misdiagnosing it. Glycogenosis type II, also known as Pompe disease (PD), is an autosomal recessive disorder caused by mutations in the *GAA* gene, localized on chromosome 17, that lead to a deficiency in the acid alpha-glucosidase enzyme (*GAA*-enzyme). This disease leads to an accumulation of glycogen in lysosomes [1,2,3], which injures the heart, leg and arm muscles, and the muscles of respiration [4]. Pompe disease is classified by age of onset, organ involvement, and severity and rate of progression. It presents two spectra of phenotypes named infantile-onset Pompe disease (IOPD) and late-onset Pompe disease (LOPD), with an incidence of 40,000–50,000 in European and American populations [5,6,7,8]. The infantile form, the most severe, is characterized by early-age onset, severe hypertrophic cardiomyopathy, marked hypotonia, macroglossia, hepatomegaly, it usually presents in the first days or weeks of life, and it has an invariably fatal outcome within one year of age. The late-onset form (juvenile or adult) is characterized by skeletal myopathy, fatigue, muscle weakness, hyperCKemia and possible respiratory failure [9,10,11].

The time of onset and phenotypes of LOPD are variable, and patients are likely to have manifestations from the fifth decade of life [4,12,13,14,15]. Mutations in the *GAA* gene, inherited in an autosomal recessive way, are very heterogeneous. They can be point mutations that can affect the functionality and stability of the protein or the splicing process, or small and large deletions and insertions [16,17,18]. Taverna et al. reported that *GAA* variants are clustered in three critical regions of the gene: exon 2, which contains the start codon exons 10 and 11, which encode for the catalytic site, and exon 14 which encodes for a highly conserved region of *GAA* protein [17]. The Pompe disease *GAA* variants database http://www.pompevariantdatabase.nl, (accessed on 26 April 2024) which was last updated in 2023, reports more than 911 disease-associated mutations [19].

Currently, enzyme replacement therapy (ERT) with *GAA*-enzyme is the first and unique treatment available for PD, and was approved in 2006 [20]. In order to achieve optimal results, ERT should be started before symptoms are evident: indeed, it is well established that early initiation of ERT in IOPD improves survival, reduces the need for mechanical ventilation, results in early independent ambulation and improves the patient’s overall quality of life [21]. In 2014, during the 208th International Workshop of the European Neuromuscular Center, the European Pompe Consortium was formed, bringing together a group of experts from 11 different European countries, all with long experience in the treatment and follow-up of patients with PD, to outline recommendations for the initiation and discontinuation of ERT in adult patients [22]. It was observed that patients receiving ERT present significant differences in their responses. These responses are heterogeneous and influenced by many factors, such as age at treatment initiation, cross-reactive immune material (CRIM) status, ACE genotype, sustained intermediate titers (SIT), muscle fiber type involvement, extent of pre-existing pathology, high and sustained antibody titers (HSAT), and ACTN genotype [23]. Cross-reactive immunological material (CRIM) is a measurement of natural *GAA* production and an important factor influencing patients’ responses to ERT. In general, CRIM-positive subjects tolerate ERT well. However, CRIM-negative subjects and some classified as CRIM-positive have a low response to ERT due to complications from the immune response to the drug [23,24,25].

To update the status of Italian subjects with both childhood onset (IOPD) and late onset (LOPD) in Italy, we present the results collected in recent years on samples that came to our attention. We evaluated the enzyme activity and genetic study of mutations in the *GAA* gene. We employed a fluorimetric test to evaluate alpha-glucosidase activity on blood spots in 2934 patients who presented with symptoms linked to PD, or for newborn screening. In the whole cohort of patients analyzed with reduced or borderline activity, genetic testing was done, and we found 39 patients with two causative mutations in *GAA* gene connected to PD, and 22 subjects with GVUS variants. These results not only underscore the critical importance of early and accurate diagnosis but also highlight the essential collaboration between clinicians and geneticists. The integration of enzyme activity screening and genetic testing represents a major advancement in our ability to identify and manage PD, ensuring patients receive timely and appropriate care. By presenting these data, we aim to emphasize the substantial progress made in the field of PD research and diagnosis in Italy, showcasing how interdisciplinary efforts are pivotal to advancing medical knowledge and patient outcomes. Our research indicates that Italian patients with Pompe disease present a diverse genetic landscape, with specific mutations in the *GAA* gene contributing to disease severity and progression. The variability in symptom onset, ranging from childhood to adulthood, underlines the complexity of genotype–phenotype correlations in this population. In addition, enzymatic and genetic testing over the years has allowed more precise diagnoses and management strategies tailored to the individual profiles of our patients. In conclusion, early diagnosis remains critical for optimizing outcomes in Pompe disease. In Italy, efforts to enhance awareness among healthcare providers have facilitated timely identification of symptoms such as muscle weakness, respiratory insufficiency, and cardiac involvement. Diagnostic modalities including enzyme assays, genetic testing, and imaging techniques play pivotal roles in confirming the disease and monitoring its progression.

### Sentinel Symptoms of PD, Diagnostic Tests, and Therapeutic Procedures

The PD challenge for clinicians begins when manifestations are found in smooth and skeletal muscle, endothelial cells and motor neurons, and in heart in the IOPD form.

The study conducted by Thurberg et al. showed that muscle cells at the start of the disease have small glycogen-filled lysosomes that grow in size and number over time. The clinical symptoms are either not visible or absent at this stage [26]. As the disease progresses, the lysosomal membranes break, thus allowing glycogen to enter into cytoplasm. Muscle damage begins and the patient experiences more visible symptoms, such as severe myopathy, which results in decreased muscle function. These can be defined as sentinel symptoms accompanied by features such as respiratory failure, high hyperCKemia (CK) values, etc. But how to begin to identify the disease?

If PD is clinically suspected, diagnosis typically involves thorough enzymatic and genetic analyses. These assessments are conducted using dry blood samples (DBS), cultured fibroblasts, or muscle tissue. In DBS samples, the presence of polymorphonuclear neutrophil-derived α-glucosidase (maltase-glucoamylase, MGA) can interfere with the accurate measurement of enzymatic activity at acidic pH levels. Techniques such as immunocapture or the competitive inhibition of MGA activity with maltose or acarbose have been utilized to eliminate the interfering effects of MGA activity in DBS extracts [27,28]. Alpha acid glucosidase has optimal activity at pH 3.7–4.5 and hydrolyzes alpha 1-4 and alpha 1-6 bonds in glucose polymers. Only if enzyme activity is below normal, borderline, or absent is a genetic assay through Sanger sequencing conducted. The study of the entire portion of the *GAA* gene for mutation research is important to the prediction of disease risk, identification of healthy carriers, and for phenotype–genotype correlations.

Because PD is an autosomal recessive disease, it is critical to investigate and know the family background after the proband is identified. Family members of an affected person have a higher-than-average chance of having the disease themselves or of being disease carriers. It is very important to verify this possibility through family screening because early diagnosis is necessary for the effective treatment and management of symptoms. It is, however, necessary that further surveys be carried out because the association between genetic factors and symptom profile is not well understood [29]. Genetic heterogeneity associated with a phenotype, and the phenotypic pleiotropy associated with a genotype, can mislead clinical presentation [30].

In the study by Salunkhe et al., the basic concepts of genetic variations and available genetic tests were explained [30]. In the study by Taverna et al., an overview of PD was presented, focusing on pathogenesis, clinical phenotypes, molecular genetics, and the role of miRNAs as potential biomarkers for PD [17]. Another group of international experts met in 2006 to discuss different aspects of PD, including genetic counseling [14]. The guideline aims to increase public awareness of this disease and introduce the broad spectrum of related symptoms in order to accelerate diagnosis and take advantage of emerging therapies. In our study, all infants with infantile-onset Pompe disease (IOPD) displayed a clinical presentation characterized by severe and progressive hypotonia (“floppy baby”- or “rag doll”-type), hypertrophic cardiomyopathy, respiratory insufficiency, and delays or regression in motor development milestones. For those with late-onset Pompe disease (LOPD), we observed elevated creatine kinase (CK) levels, feeding difficulties, and progressive motor dysfunction such as difficulty raising the upper limbs, climbing stairs, and rising from a low chair, often leading to respiratory failure. Additionally, moderate hepatomegaly was noted in a few cases. Some subjects had previous muscle biopsy results available.

Various other symptoms were also identified to varying extents, including dysphagia, osteoporosis, sleep apnea, small-fiber neuropathy, impaired gastric function, urinary tract involvement (monitored by Glucose Tetrasaccharide (Glc4) Urine Testing), pain, and fatigue. These findings underscore the systemic nature of Pompe disease and emphasize symptoms that unfortunately may not have been promptly recognized as related to the disease. Regarding the genetic analysis of our samples, we did not identify a clear genotype–phenotype correlation. We believe that other factors beyond genetic mutations may influence the clinical manifestations of Pompe disease, factors that are not yet fully understood. Therefore, all subjects investigated in our study represent hypothetical Pompe disease patients. This comprehensive characterization highlights the diverse clinical spectrum and challenges in diagnosing Pompe disease, emphasizing the critical need for increased awareness and early intervention strategies to improve patient outcomes.

The interest of the scientific community in the treatment of inherited diseases has focused on the continuous development of drugs and therapies that use molecules to modify gene expression, and correct or compensate for an abnormal phenotype. In particular, strategies include resolution of disease enzyme deficiency (enzyme replacement therapy-ERT), replacement of the gene encoding the deficient enzyme (gene therapy), and cell dispensing that produces the deficient protein (hematopoietic stem cell transplantation) [31,32,33,34]. Other therapeutic strategies include administering drugs that enhance the activity of mutated proteins (enzyme-enhancement therapy), or reducing substrate biosynthesis, the accumulation of which results in toxicity (substrate reduction therapy) [35,36,37,38] (Figure 1).

To date, the only specific treatment available for PD patients is enzyme replacement therapy, which aims to arrest the natural course of the disease [28,39,40]. Both the success and limitations of ERT have provided new insights into the disease’s pathophysiology and motivated the scientific community to develop the next generation of therapies that will be introduced into clinical procedure [21,41,42,43].

The replacement enzyme is produced biotechnologically and administered intravenously. ERT with *GAA*-enzyme significantly prolonged the survival of IOPD patients, with marked improvements in symptomatology. In LOPD, it is only effective in slowing the progression of the disease. Consequently, this treatment is more effective in the IOPD than the LOPD form. In addition, systemic administration through the blood does not effectively target all affected organs [44,45]. However, it is unquestionable that ERT has provided many benefits, adding years and quality of life to PD patients, but it is equally clear that there is a need to improve and optimize the current drugs, which still have several limitations [31,40,46,47]. In recent years, there has been a significant increase in the number of research projects, both in vitro and in vivo mouse models, with the goal of improving the PD patient’s treatment [48]. In our study of 39 patients diagnosed with late-onset Pompe disease (LOPD), Italian clinicians monitored the effects of enzyme replacement therapy (ERT) on their motor function over time. Upon initiating therapy, patients generally exhibited a partial improvement in their motor abilities within the first year. This initial phase of treatment showed encouraging signs, with noticeable gains in muscle strength and mobility. As therapy continued beyond the first year, the trajectory of disease progression changed. Rather than continuing to show significant improvements, the patients’ motor functions stabilized. This stabilization indicates that, while the rapid initial improvements tapered off, the therapy played a crucial role in maintaining the gains achieved during the first year and in slowing the overall progression of LOPD. These findings suggest that enzyme replacement therapy provides substantial early benefits in terms of motor function, and while it may not lead to continuous improvement, it is effective in preventing further decline. This highlights the importance of early and sustained treatment in managing LOPD and improving patients’ quality of life.

## 2. Results

We evaluated the enzyme activity of 2934 subjects, 1670 males and 1264 females, divided by age. Mean age was 31 years at time of analysis: 165 subjects aged 0 to 1 years old suspected to have IOPD, and 2769 subjects over 1 year old suspected to have LOPD. The typical clinical features reported in Toscano et al. were found in all our examined subjects [49]. Due to the importance of scientific health research to date, a study of subjects from different hospital departments was conducted to delineate the clinical phenomenon in an observational and analytical view. DBS received with suspected PD came from: 1380 neurology (47.03%), 40 metabolic diseases (1.36%), 473 pediatrics (16.12%), 116 pneumology (3.95%), 177 internal medicine (6.03%), 345 rheumatology (11.76%), 322 from other departments such as cardiology, anesthesia and reanimation, hematology, and medical genetics (10.97%), and 81 from unspecified departments (2.76%) (Figure 2).

The first step was to assess the outcome of enzyme activity. Of the 2934 individuals studied by DBS, 2489 showed normal acid α-glucosidase activity (≥6 nmol/mL/h), 105 individuals had no or reduced activity of the enzyme (≤3 nmol/mL/h), and 340 had an enzyme activity between 3 and 6 nmol/mL/h.

In subjects with enzyme activity <6 nmol/mL/h, we performed genetic analysis on 445 subjects to research and identify any change in allelic variants of the *GAA* gene. In 105 subjects with none or severely reduced enzyme activity, we identified 39 patients who had either homozygosity for one mutation, or compound heterozygosity for two mutations in the *GAA* gene, known in the literature to be responsible for PD. Segregation analysis in the patients’ family with Sanger sequencing confirmed that, in the 31 cases of compound heterozygote, mutations were on both homologous chromosomes. This confirmed the diagnosis of PD from a genetic point of view for all patients analyzed.

In the remaining 51 of the 105 subjects were false positives in which the enzymatic analysis showed values below normal, but no causative mutations in the *GAA* gene were found in the genetic analysis. The remaining 15 out of the 105 subjects had a causative mutation in simple heterozygosity. Therefore, the reduced enzyme activity could be due to a leukopenia condition and/or improper storage/transport of the sample.

The most common mutation reported in our analyzed samples was the pathogenic c.-32-13 T>G variant (homozygous or with a second pathogenic allele), located in intron 1 documented in the literature as a causative mutation of PD. All variants found in our molecular study associated with the c.-32-13 T>G mutation were reported in the bibliographies and public databases of PD mutations (Pompe variant database, clinvar). Specifically, 25 of 39 patients reported this causative mutation. Of these 25, 5 patients had the mutation in homozygosity and 20 patients in compound heterozygosity (Table 1).

In addition, the study was further investigated by conducting MLPA in patients who had only one mutation on Sanger sequencing analysis. The assay was performed in 149 subjects. This method revealed in four subjects the presence of exonic deletions in the *GAA* gene causative of PD in association with other causative point mutations. In detail, the deletions found in our patients were located in exon 9, exon 17, exon 18, and exon 19, respectively, in the first four cases.

In the other four, three investigations were conducted for a family segregation study in which, in addition to the deletion, benign polymorphisms were found. One patient showed a deletion in exon 12.

Four of these eight subjects had the deletion in association with other causative point mutations, as shown in Table 1, and for the remaining four no additional mutations in the *GAA* gene were found.

In our study, we identified a total of 39 subjects with PD, including 27 with two classic mutations in compound heterozygosity, 8 with mutations in homozygosity, and 4 with one causative mutation associated with a large deletion. Specifically, as shown in Figure 3, these subjects came from the following departments: 17 from neurology (43.6%), 4 from rheumatology (10.3%), 7 from pediatrics (17.9%), 3 from metabolic diseases (7.7%), and 8 from other departments (20.5%).

Furthermore, our study identified five allelic variants in the sequence of the *GAA* gene (c.547-67C>G, c.547-39T>G, R437H, L641V, and L705P) that had not been previously described in the literature. In the entire cohort of patients analyzed with enzyme activity between 3 and 6 nmol/mL/h, we also detected 22 patients with variants of uncertain significance (GVUS) mutations in heterozygosity (Figure 4).

In our study, we investigated all genetic polymorphisms as a possible factor contributing to PD. These were classified according to the type of genetic polymorphism investigated and whether the result favored or did not favor association with PD. As asserted by Ravaglia et al., genetic variations impacting muscle structure and metabolism could potentially influence the phenotype of metabolic myopathies [50].

We found in our subjects frequently benign polymorphisms, as reported in databases of PD-associated mutations and annotated as benign variants, and which were thus unrelated to PD.

These statistical results collected over a recent period geographically cover the entire Italian demographic area. Therefore, this clinical study plays an important role in knowing and being able to identify typical signs and symptoms of PD.

## 3. Discussion

The diagnosis of PD poses challenges due to its diverse clinical presentations among patients, the rarity of the condition, and the complexities involved in recognizing its symptoms. Consequently, it often remains underdiagnosed. This article aimed to highlight the increasing number of PD Italian cases identified through dried blood spot (DBS) testing in recent years. Additionally, this study incorporated the assessment of quantitative variations in the *GAA* gene using the multiple ligation-dependent probe amplification (MLPA) assay to enhance diagnostic accuracy. As discussed, for autosomal recessive diseases, the presence of two causative mutations in compound heterozygosity or homozygosity defines an individual with PD. To determine whether the two mutations are in cis or trans positions, a genetic investigation was conducted within the family to see how the mutations were transmitted in their chromosomes and thus inherited by the individual under study. The cis configuration occurs when two mutant alleles are on the same homologous chromosome, while the trans configuration occurs when the two mutant alleles are on different homologous chromosomes.

Our study concluded that Italian patients exhibited a characteristic genetic profile similar to that of the broader European population, with the c.-32-13T>G variant being the most prevalent. Genetic analysis also indicated that the functional significance and potential interrelationships of numerous benign polymorphisms, as well as other genetic variants of uncertain significance (GVUS), are still relatively unknown. We can conclude that our results allow us to set the cutoff for enzyme activity at <3 nmol/mL/h for the identified cases of PD.

For MLPA, large deletions make up only 1.5% of known variants, and the deletion of exon 18 was a common variant in Caucasian patients. A more systematic approach, as previously described in our study, could lead to the identification of new variants associated with PD and allow for a definitive diagnosis of patients. The implementation of MLPA should be considered to complete the genetic analysis of the most challenging cases, where there is a strong clinical suspicion supported by signs and symptoms associated with pathological enzyme activity values. Given the extensive heterogeneity of the *GAA* gene, both in enzymatic and genetic characteristic terms, additional efforts are required to investigate the potential influence of environmental and epigenetic factors on phenotype modulation. Furthermore, for improved patient classification, future studies should be conducted through close collaboration between neuropsychiatry divisions and genetic laboratories. In summary, prompt diagnosis is crucial for maximizing outcomes of Pompe disease. In Italy, initiatives to raise awareness among healthcare providers have facilitated timely recognition of symptoms such as muscle weakness, respiratory insufficiency, and cardiac involvement. Diagnostic methods such as enzyme assays, genetic testing, and imaging techniques are essential for confirming the disease and tracking its progression. This information could be useful for developing new biomarkers for this disease. The advent of enzyme replacement therapy (ERT) has revolutionized the management of Pompe disease, offering a targeted approach to supplementing deficient *GAA* enzyme activity. Italian patients have benefited from access to ERT, which aims to alleviate symptoms, improve muscle function, and enhance quality of life. Ongoing research explores novel therapeutic avenues, including gene therapy and pharmacological chaperones, to further optimize treatment outcomes and address challenges such as immune response and sustained efficacy. In conclusion, Pompe disease presents complex clinical and genetic landscapes among Italian patients, underscoring the importance of ongoing research and multidisciplinary collaboration in enhancing diagnostic accuracy, therapeutic efficacy, and overall patient outcomes. Continued efforts are essential to improve disease-management strategies, expand treatment options, and ultimately improve the lives of individuals living with Pompe disease, both in Italy and globally.

## 4. Materials and Methods

### 4.1. Patients

Our data collected in recent years allowed the identification of patients with a pathological value of alpha-acid glucosidase enzyme activity and the presence of *GAA* mutations. The biochemical and genetic investigations that led to PD diagnosis were performed at the Centre for Research and Diagnosis of Lysosomal Storage Disorders at IRIB-CNR in Palermo, and were approved by the Hospital Ethics Committee of the University of Palermo (Sicily, Italy). Signed informed consent was obtained from patients. For each patient, clinical history and laboratory data were collected in a case report form.

### 4.2. Sample Collection

The peripheral blood of the patient was collected using EDTA as an anticoagulant, and dried in a specific adsorbent paper spot (DBS). Blood was dropped and allowed to dry for 24 h at room temperature before being sent (with 1–2 days of transport) to the centralized analytical laboratory in Palermo. Sample storage after use was conducted at a temperature of 4 °C. The same samples from analyzed subjects were used for multiple analyses (enzymatic and genetic assay).

### 4.3. Enzymatic Assay

The analysis of *GAA* enzyme activity was conducted on dried blood spots. The method used was described by Chamoles et al. [27], with modifications to improve the precision of the enzymatic activity measurement. DBS samples were examined by fluorometry techniques. The fluorometric method uses the substrate 4-methylumbelliferyl-α-D-glucoside [51]. Genetic analysis was performed in patients with *GAA* activity less than 6 nmol/mL/h.

### 4.4. DNA Extraction

DNA extraction was performed by the whole DBS spot, corresponding to approximately 3 µL of blood per punch, followed by the addition of 310 microliters of the digestion buffer (Buffer G2—Qiagen GmbH, Hilden, Germany) and 15 microliters of the Proteinase K solution (Qiagen GmbH lot No. 175025821) per each individual sample. DNA extraction was performed using a Qiagen EZ1 advanced XL (QIAGEN Redwood City-1001 Marshall Street-Redwood City, CA, USA) automatic extractor in combination with the EZ1 DNA investigator kit.

### 4.5. Genetic Analysis-PCR, Sanger Sequencing, and MPLA

Eleven pairs of primers were designed to analyze twenty target regions containing the twenty exons of the *GAA* gene. PCR products were purified and analyzed by Sanger sequencing, using an automated DNA sequencer (Eppendorf-Donau-City-Straße 11-13, 3. OG 1220 Wien, Austria) to identify mutations. Multiplex ligation-dependent probe amplification (MLPA) is a relatively new diagnostic method to evaluate large deletions and duplications undetectable by *GAA* gene sequencing, using a very small amount of DNA. This analysis was conducted using the kit “SALSA MLPA Probemix P453 *GAA*” MRC-Holland. Nomenclature of the variants in this article refer to NM_000152.5(*GAA*) (https://www.ncbi.nlm.nih.gov/clinvar, accessed on 1 August 2024).

## 5. Conclusions

To satisfy the ongoing need for knowledge and to promote research, a study was conducted in our laboratories to describe biochemical findings and the mutation spectrum of the *GAA* gene for PD patients. Although this metabolic disorder is caused by a unique enzyme deficiency, it appears with a wide range of clinical phenotypes and with considerable variability in progression speed, degree of organ involvement, and disease severity. This is related to the number of known pathogenic mutations, their kind (nonsense, missense, framesahift), and their mutual interaction with the rest of the genome [52,53,54,55,56,57]. This report presents a study of Italian subjects with PD, highlighting the importance of knowledge of the disease so that the clinician can recognize it and request enzyme and genetic counselling to avoid misdiagnosis.

This report presents a work on Pompe disease, highlighting the importance of genetic counseling and enzyme testing to avoid misdiagnosis. Diagnostic delay or misdiagnosis in a patient with PD signs and symptoms are the most critical points in determining the evolution of the patient’s clinical and therapeutic history. The right strategic approach is the most effective solution to improve the course of the disease and patient’s outcome. In addition, the possibility of extending the genetic study to family members of identified affected probands promotes not only early diagnosis but also timely therapeutic intervention.

## Figures and Tables

**Figure 1 ijms-25-09139-f001:**
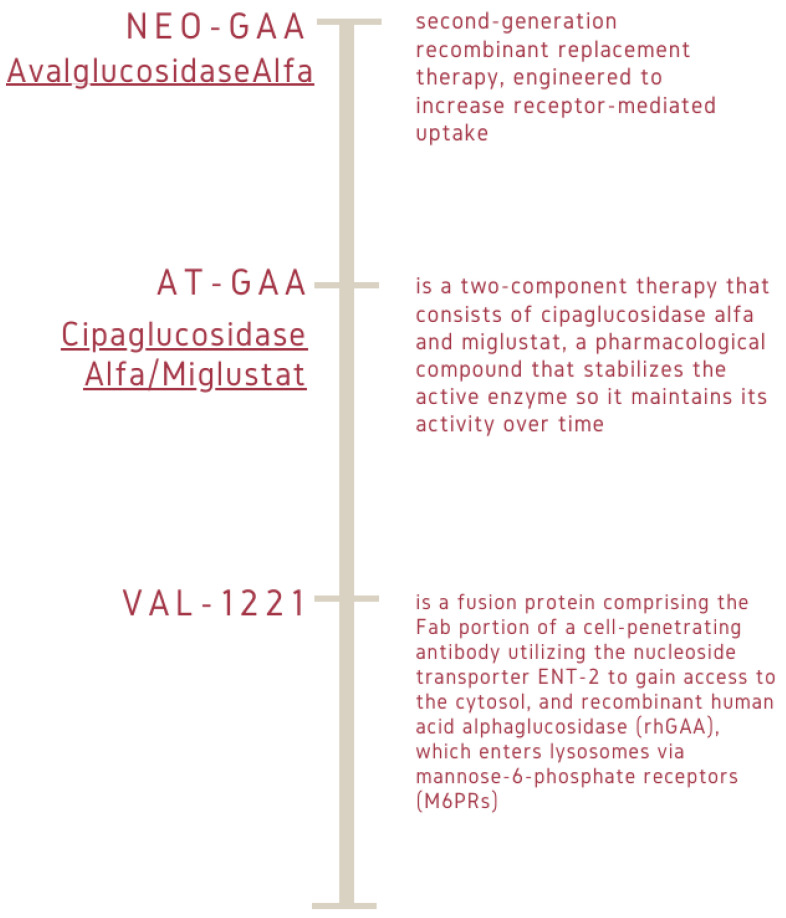
Different types of molecular drugs involved in gene-expression regulation. Avalglucosidase alpha, or Neo*GAA*, is a drug for enzyme replacement therapy, specifically designed for Pompe disease. Avalglucosidase alpha is a recombinant form of *GAA* that restores deficient enzyme levels. First developed by Sanofi Genzyme, avalglucosidase alpha is a chemically modified version of alglucosidase alpha, where synthetic bis-phosphorylated oligosaccharides were attached to the structure to improve cellular uptake of the drug and better muscle targeting. Pombiliti + Opfolda (cipaglucosidase alpha/miglustat), formerly known as AT-*GAA*, is an approved combination therapy for adults with late-onset Pompe disease, or LOPD. VAL-1221 is a fusion protein comprising the Fab portion of a cell-penetrating antibody utilizing the nucleoside transporter ENT-2 to gain access to the cytosol, and recombinant human acid alpha glucosidase (rh*GAA*), which enters lysosomes via mannose-6- phosphate receptors (M6PRs).

**Figure 2 ijms-25-09139-f002:**
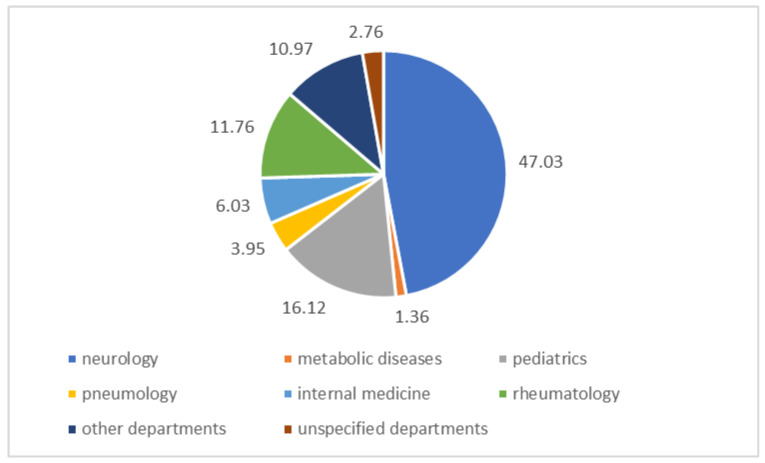
Sources of the 2934 subjects studied.

**Figure 3 ijms-25-09139-f003:**
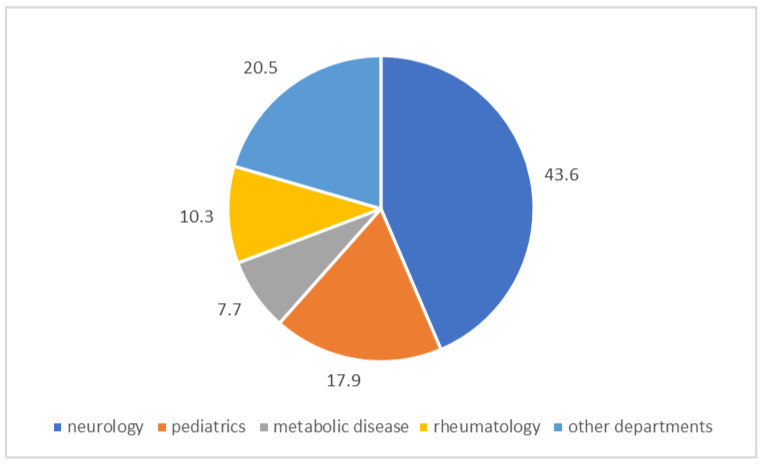
Sources of the 39 PD subjects.

**Figure 4 ijms-25-09139-f004:**
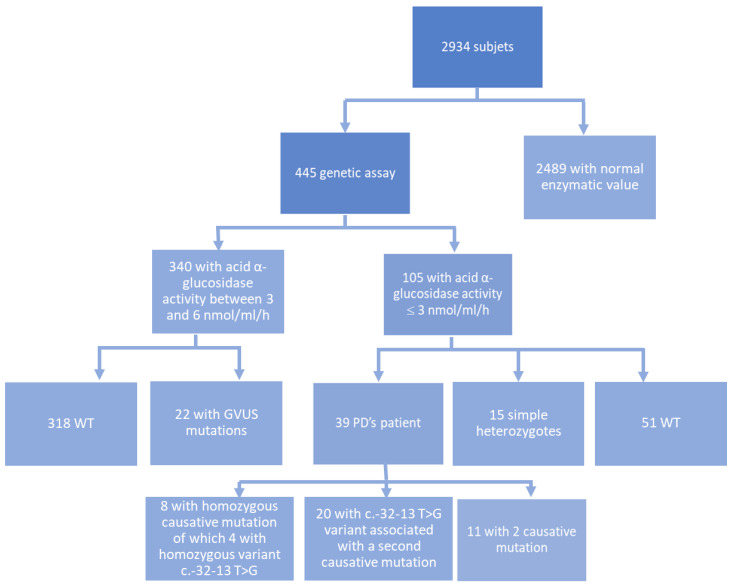
Flowchart diagram: reference value of activity enzyme is ≥6 nmol/mL/h.

**Table 1 ijms-25-09139-t001:** Genetic analysis information of 39 Italian patients with late-onset glycogen storage disease. Reference value of activity enzyme is (≥6 nmol/mL/h).

Subject	Age	Family History	Gender F Female M Male	α-Glucosidase Activity (DBS)	Variant 1	Variant 2
#1	57	yes	F	2.1	c.-32-13T>G	c.1655 T>C L552P
#2	54	yes	M	0.7	c.-32-13T>G	c.1655 T>C L552P
#3	38	no	M	0.8	c.-32-13T>G	c.2431 delC
#4	45	no	M	0.6	c.-32-13T>G	c.2242dupG
#5	21	no	M	1.6	c.-32-13T>G	c.1655 T>C L552P
#6	2	no	M	0.6	c.-32-13T>G	c.1927 G>A G643R
#7	NS	no	M	0.8	c.-32-13T>G	c.1465 G>A D489N
#8	10	no	M	1.4	c.-32-13T>G	c.2238 G>A W746
#9	54	no	M	1.5	c.-32-13T>G	c.2560 C>T R854
#10	49	no	F	1.3	c.-32-13T>G	c.1210 G>A D404N
#11	63	yes	M	0.9	c.-32-13T>G	c.526 delG E176R fs45
#12	NS	no	NS	0.9	c.-32-13T>G	c.784 G>A E262K
#13	82	yes	M	1	c.-32-13T>G	c.784 G>A E262K
#14	33	yes	F	1.3	c.-32-13T>G	c.525 delT E176R fs45
#15	9	yes	M	1.5	c.-32-13T>G	Del ex18
#16	35	no	F	5.4	c.670 C>T	c.861C>T
#17	5	si	M	1.1	c.-32-13T>G homozygous	
#18	2 month	no	M	0.4	c.1465 G>A D489N	c.2238 G>A W746
#19	78	no	M	1.5	c.-32-13T>G homozygous	
#20	24 days	no	M	0	c. 1565 C>G P522R homozygous	
#21	58	no	M	0.8	c.-32-13T>G homozygous	
#22	59	yes	M	0.7	c.-32-13T>G homozygous	
#23	52	yes	M	1.6	c.-32-13T>G homozygous	
#24	3 month	no	M	0.9	c.526 delG E176R fs45	c.2104 C>T R702C
#25	11 month	no	F	0.5	c.1655 T>C L552P	c.2161 ins G E721G fs16
#26	1 month	no	F	0.9	c.784 G>A E262K	Deletion on exon 9 and 19
#27	42	yes	M	5	c.784 G>A E262K	Deletion on exon 9 and 17
#28	7	no	F	0.4	c.784 G>A E262K	C.1979 G>A R660H
#29	5 month	no	M	0.4	c.1327-18 A>G	c.1655 T>C L552P
#30	7	no	F	1.1	c.784 G>A E262K	c.1979 G>A R660H
#31	12	no	F	0.5	c.2012 T>A M671K homozygous	
#32	6 mouth	no	M	0.8	c.1437+2 T>C	c.1655 T>C L552P
#33	53	no	M	20.3	c.-32-13T>G	c.1551 +1 G>C
#34	47	yes	F	5.4	c.-32-13T>G	c.655 G>A G219R
#35	15	no	M	4.2	c.2332-40 C>G homozygous	
#36	70	yes	F	3.3	c.-32-13T>G	c.1551+1 G>C etero
#37	45	no	M	0.6	c.-32-13T>G	c.2242dupG E748G
#38	24	yes	M	3.3	c.-32-13T>G	Deletion on exon 18
#39	54	yes	F	0.6	c.118 C>T R40	c.2647-7 G>A

## Data Availability

The raw data supporting the conclusions of this article will be made available by the authors on request.

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
