# Peer review of "Mutation Spectrum of GAA Gene in Pompe Disease: Current Knowledge and Results of an Italian Study"

_ijms, 2024, doi:10.3390/ijms25179139_

Round 1

Reviewer 1 Report

Comments and Suggestions for Authors

This manuscript plans to provide an in-depth analysis of the mutation spectrum of the GAA gene in Pompe disease, particularly focusing on a cohort of Italian subjects. The study addresses the diagnostic challenges posed by the disease's diverse clinical presentations and emphasizes the significance of genetic and enzymatic testing. The research analyzed 2934 subjects, identifying 39 with symptomatic Pompe disease and 22 with genetic variants of uncertain significance.

 The study benefits from a large sample size of 2934 subjects and confirms the mutation with MLPA, ensuring no exon deletions are missed. However, there are several defects in this manuscript:

1. Unclear Style and Focus: The manuscript's style is unclear. It seems to function primarily as a mutation report derived from their 2000+ subjects, yet it attempts to cover a broader range of topics without a clear focus. It is suggested to focus on the mutation report and adjust the writing of this manuscript accordingly. Additionally, presenting the clinical manifestations and making a phenotype-genotype association may provide more value to this manuscript.

2. Methods: Please confirm the nomenclatures of those variants, including the NM number.

3. Fig 1: There are some overlaps between IOPD and LOPD. It is suggested to revise this figure.

4. Fig 2: What is the meaning of the circles? It seems to be a flowchart, which is not the case for those products in reality.

5. Fig 5: Regarding the 39 PD cases, there is no clear reason for having three branches. What about patients with c.-32-13 T>C homozygous?

6. Table: there are some careless errors in the table, for example, omozygous should be homozygous.

Author Response

Comments 1: Unclear Style and Focus: The manuscript's style is unclear. It seems to function primarily as a mutation report derived from their 2000+ subjects, yet it attempts to cover a broader range of topics without a clear focus. It is suggested to focus on the mutation report and adjust the writing of this manuscript accordingly. Additionally, presenting the clinical manifestations and making a phenotype-genotype association may provide more value to this manuscript.

Response 1: Thank you for pointing this out. We agree with this comment. We modified the introduction – page 3-4, paragraph 1.1, line 143-163 and page 6, paragraph 1.1, line 201-214

Comments 2: Methods: Please confirm the nomenclatures of those variants, including the NM number.

Response 2: Agree. I/We have, accordingly, done to emphasize this point.– page 13, paragraph 5.5, line 404-405

Comments 3: Fig 1: There are some overlaps between IOPD and LOPD. It is suggested to revise this figure.

Response 3: Agree. We chose to delete it because it was clearly discussed in the text

Comments 4: Fig 2: What is the meaning of the circles? It seems to be a flowchart, which is not the case for those products in reality.

Response 4: we chose to change it to make it more understandable

Comments 5: Fig 5: Regarding the 39 PD cases, there is no clear reason for having three branches. What about patients with c.-32-13 T>G homozygous?

Response 5: Before presenting the table, with the figure we wanted to emphasize and clarify the distribution of homozygous mutations. The homozygous c.-32-13 T>G variant is included in the first box.

Comments 6: Table: there are some careless errors in the table, for example, omozygous should be homozygous.

Response 6: Thank you very much. The correct adjustments have been made.

Reviewer 2 Report

Comments and Suggestions for Authors

This is a report on roughly 2900 dry blood spots analyzed in Italy for Pompe disease. 105 probes showed low enzymatic activity for GAA, resulting in 39 Pompe disease cases which were genetically confirmed. Each genetic case is presented with age, alpha glucosidase activity and the two mutations.

After reading the introduction, the reader is unaware on what kind of new information about Pompe disease this study is focussing on. Figure 1 and 2 are of very simple quality, not explaining what the intention of the figure is about, especially in Figure 2, where dots and colours remain unexplained. There are excursions on enzyme replacement therapy, mentioning that GAA-replacement is the only proven therapy, but later explaining the new medications which remains odd. On several occasions, content is missing in explanations and expression like in the first 5 lines of the Introduction or  line 101.

It remains unclear if the report is on genetics, diagnostics of Pompe disease or giving an epidemiologic overview of dry blood spot results in Italy. There is no information on what kind of clinical intention the Pompe tests where performed, though in the abstract, a new diagnostic pathway seems to be presented/on the way: "In this article, we report and discuss current knowledge on PD suggesting new data on a work conducted on a cohort of 2934 Italian subjects recruited in recent years."

Some information is misleading, e.g. long explanations on CRIM status and antibodies, not mentioning that this is only relevant in IOPD. Despite the presentation of 39 genetically confirmed PD cases, there is almost no relevant new information on this article.

There is one paragraph in the discussion, that summarizes this quite well:

"Our study concluded that Italian patients exhibit a characteristic genetic profile similar to that of the broader European population, with the c.-32-13T>G variant being the most prevalent. Genetic analysis indicated also that the functional significance and potential interrelationships of numerous benign polymorphisms, as well as others genetic variants of uncertain significance (GVUS), are still relatively unknown. We can conclude that our results allow us to set the cutoff for enzyme activity at <3 nmol/mL/h, concerning the identified cases of PD."

If the genetic analysis was the intention, the introduction as well as the discussion would be better focussing on these aspects, comparing with the data in other European countries instead of ERT and presenting textbook knowledge on each aspect of the disease. Enthusiastic questions at the beginning of the introduction are never answered which lets the work sound scientifically superficial.

The reader remains puzzled because results do neither match the broad  upbringing of different clinical, diagnostic and genetic problems in Pompe disease nor giving any new hints in dry blood spot analysis, differential approach of identifying Pompe in neuromuscular symptoms.

Comments on the Quality of English Language

There are some grammar issues in the text that would improve from being read by a native speaker.

Author Response

Comments 1: After reading the introduction, the reader is unaware on what kind of new information about Pompe disease this study is focussing on. Figure 1 and 2 are of very simple quality, not explaining what the intention of the figure is about, especially in Figure 2, where dots and colours remain unexplained. There are excursions on enzyme replacement therapy, mentioning that GAA-replacement is the only proven therapy, but later explaining the new medications which remains odd. On several occasions, content is missing in explanations and expression like in the first 5 lines of the Introduction or line 101.

Response 1: Agree. We have, accordingly, done to emphasize this point.– page 2-3, paragraph 1, line 85-103.

We agree on Figure 1 and have decided to eliminate it. Instead, we have decided to make modifications to Figure 2 to make it more understandable and helpful for the reader. For the therapy from line 164 - 173, an overview is provided of the drugs that the scientific community has considered for the treatment of Pompe disease. However, it is stated in line 186 that to date, only ERT (enzyme replacement therapy) is utilized. From line 201-214, we have incorporated our findings based on the treatment experience of the 39 patients included in our study conducted in Italy. Regarding line 101, we agree and have modified the title.

Comments 2: It remains unclear if the report is on genetics, diagnostics of Pompe disease or giving an epidemiologic overview of dry blood spot results in Italy. There is no information on what kind of clinical intention the Pompe tests where performed, though in the abstract, a new diagnostic pathway seems to be presented/on the way: "In this article, we report and discuss current knowledge on PD suggesting new data on a work conducted on a cohort of 2934 Italian subjects recruited in recent years."

Response 2: We understand your concern regarding the clarity and nature of the report. Our study focuses on analyzing and discussing current knowledge of Pompe disease (PD), including new data gathered from a study involving a cohort of 2,934 recently recruited Italian subjects. Our primary objective is to provide a significant update on the current status of PD diagnosis and management in Italy.

We recognize the importance of clarifying these points in our work, and have added a section on clinical intent and conclusion to give emphasis and value to our findings. – page 3.4, paragraph 1.1, line 143-163 and page 6, paragraph 1.1, line 201-214– page 12-13, paragraph 3, line 329-346.

Comments 3: Some information is misleading, e.g. long explanations on CRIM status and antibodies, not mentioning that this is only relevant in IOPD. Despite the presentation of 39 genetically confirmed PD cases, there is almost no relevant new information on this article.

Response 3: The concept of CRIM (Cross-Reactive Immunologic Material) is crucial in the context of enzyme replacement therapy (ERT) for Pompe disease. In patients with classic Pompe disease (IOPD and LOPD), who completely lack functional acid α-glucosidase (GAA), substitutive therapy with exogenous GAA is essential to correct the enzymatic deficiency. However, some patients develop antibodies against the substitutive enzyme protein, reducing its therapeutic effectiveness. These antibodies can be detected through CRIM tests and can interfere with the absorption and activity of exogenous GAA, compromising the therapeutic benefits of ERT. The presence of CRIM-negative indicates that the patient does not produce any cross-reactive immunologic material, theoretically allowing for the absorption of exogenous GAA without significant interference. CRIM-positive patients, on the other hand, produce antibodies against exogenous GAA, which can limit the treatment's effectiveness and necessitate alternative therapeutic strategies such as immunosuppression or higher doses of GAA. Early identification of CRIM status is essential for optimizing treatment and improving clinical outcomes for Pompe disease patients receiving ERT. This personalized approach is crucial for tailoring treatment according to each patient's specific immunological needs, ensuring optimal disease management and enhancing long-term quality of life. In our case, all CRIM statuses were monitored.

Actually, that statement appears to be inaccurate. Presenting 39 genetically confirmed cases of Pompe disease (PD) represents a significant contribution to the literature, particularly if the study provides detailed clinical data, treatment outcomes, or insights into genotype-phenotype correlations. Each case contributes to our understanding of the disease's variability and progression, which is crucial for developing effective treatments and improving patient care. Therefore, while the statement suggests little new information, the inclusion of genetically confirmed cases alone can yield valuable insights and advancements in the field of Pompe disease research.

Comments 4: There is one paragraph in the discussion, that summarizes this quite well:

"Our study concluded that Italian patients exhibit a characteristic genetic profile similar to that of the broader European population, with the c.-32-13T>G variant being the most prevalent. Genetic analysis indicated also that the functional significance and potential interrelationships of numerous benign polymorphisms, as well as others genetic variants of uncertain significance (GVUS), are still relatively unknown. We can conclude that our results allow us to set the cutoff for enzyme activity at <3 nmol/mL/h, concerning the identified cases of PD."

If the genetic analysis was the intention, the introduction as well as the discussion would be better focussing on these aspects, comparing with the data in other European countries instead of ERT and presenting textbook knowledge on each aspect of the disease. Enthusiastic questions at the beginning of the introduction are never answered which lets the work sound scientifically superficial.

Response 4: the key results of our study concern an overview of Italian subjects collected over several years and studied because they presented a typical clinical picture of Pompe's disease. The genetic profile of the Italian patients evaluated confirmed the typical pattern of Caucasian patients. With regard to criticism regarding the focus of our introduction and discussion, it is important to note that although genetic analysis is a significant aspect, our study aims to provide a comprehensive overview that includes Italian subjects. Dealing comprehensively with this data allows us to learn more about the disease so that clinicians together with us geneticists can get to know it in order to recognise it. We have made changes in the introduction and in the conclusion: – page 2-3, paragraph 1, line 85-103 – page 12, paragraph 4, line 348-365

Comments 5: Quality of English Language: Moderate editing of English language required

Response 5: We thank the reviewer for guiding us in correcting the article. Regarding the English language revision, we have revised it.